



# Finely laminated Arctic mixed-phase clouds occur frequently and are correlated with snow

Emily M. McCullough[1,*], Robin Wing[2], and James R. Drummond[1]

[1]Department of Physics and Atmospheric Science, Dalhousie University, 6310 Coburg Rd., PO Box 15000, Halifax, NS, B3H 4R2, Canada
[2]LATMOS/IPSL, OVSQ, Sorbonne Universités, CNRS, Paris, France

*Correspondence to:* Emily McCullough (e.mccullough@dal.ca)

**Abstract.**

Finely laminated (multi-layer) clouds, which are strongly correlated with precipitation events, have been detected in 3.5 years of high resolution measurements of Arctic mixed-phase clouds using the Canadian Network for the Detection of Atmospheric Composition Change (CANDAC) Rayleigh-Mie-Raman lidar located at Eureka, Nunavut (79.6° N, 85.6° W).

Laminated clouds occur on 52% of days with 24 h measurement coverage from 0-5 km altitude, and on 62% of cloudy interpretable days. There is an average of 70 laminated cloud days detected per year, with no full year having fewer than 52 detections. Given CRL does not measure on all days of the year, it is probable that the true occurence frequency of laminated clouds at Eureka is much higher.

A study was conducted using local weather reports from the nearby Environment and Climate Change Canada (ECCC)
weather station. Days with laminated clouds are strongly correlated with snow precipitation, while days with non-laminated clouds and clear sky days are moderately anti-correlated with snow precipitation.

## 1   Introduction

Arctic clouds generally warm the surface by trapping and re-emitting upwelling infrared radiation, except in summer when they contribute a slight cooling by emitting more radiation to space than they reflect back to the surface (Nott and Duck, 2011;
Intieri et al., 2002). Shupe et al. (2011) describe a 70% annual average cloud fraction at Eureka, with cloud fractions of 50% (in May) to 80% (September through March), measured during a 2005-2009 study period. High interannual variability was noted in the monthly distribution of cloud occurrence, but the total annual cloud occurance was constant to within $\pm 1\%$. During the Arctic polar night, in the absence of incoming solar radiation, clouds can dominate the radiation budget, so understanding their radiative impact is essential (e.g. Noel et al. (2006); Cess et al. (1990, 1996); Platt et al. (1998)). Given the sensitivity of the
atmospheric radiation balance to cloud particle phase, warm clouds (exclusively liquid), cold clouds (exclusively ice, but with multiple formation processes including homogenous and heterogeneous freezing), and the rather more complex mixed-phase clouds, should all be characterized.

Mixed-phase clouds are ubiquitous in the Arctic. Over the Arctic as a whole, satellite measurements show occurance frequency from 30% during winter to 50% during the rest of the year, but the spatial distribution is not uniform. Specific locations,

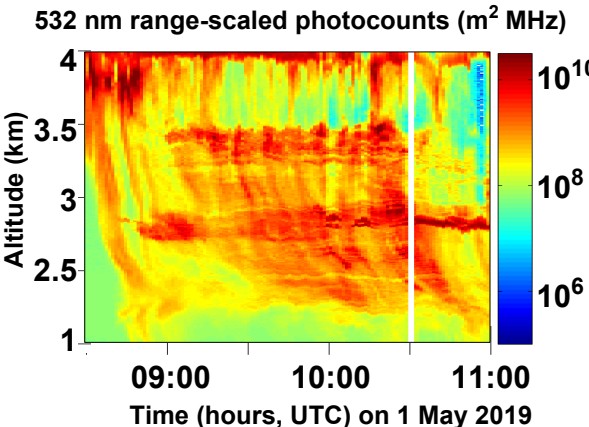

**Figure 1.** Thin laminated layers within an Arctic cloud with fall streaks also visible. 532 nm range-scaled counts from the CRL lidar at Eureka, Nunavut, on 1 May 2019. This is an excerpt of the plot in Fig. 3d.

e.g. Svalbard, show up to 90% occurance at low altitudes for parts of the year (Mioche et al., 2015). Multi-year ground- and ship-based measurements in Alaska show 41% average occurrance frequency throughout the year (59% frequency when clouds were present), increasing to 70% in the fall. There is a 25% annual average fraction of mixed-phase clouds at Eureka (Shupe, 2011). Therefore, mixed-phase clouds are an important component of the Arctic radiative budget.

Mixed-phase clouds involve a complex interaction of three phases of water (vapour, liquid, and ice) coexisting in the same cloud, and are particularly challenging to model. Models in the 1980s and 1990s were not capable of adequately predicting particle size spectra within clouds, and those which predicted cloud height, depth and water content were hoping to benefit from further measurements which parameterize the cloud optical properties in terms of liquid and ice water content (Heymsfield and Platt, 1984; Sun and Shine, 1995). Numerous studies measuring liquid water path (LWP) have since been undertaken (e.g.

Turner (2005); Bühl et al. (2016)). Mixed phase clouds are now known to contain little liquid water, but this small amount, in all seasons, has a dominant effect on local cloud-surface radiative interactions (Luke et al., 2010; Shupe and Intieri, 2004), top-of-atmosphere radiation budgets (Korolev et al. (2017), after Dong and Mace (2003) and Zuidema et al. (2005)) and global climate (Tan et al., 2019).

Numerous shortcomings in climate and weather models and reanalysis products have been attributed to an inadequate rep-

resentation of mixed phase clouds, and/or the presence of supercooled liquid within these clouds. Tan et al. (2019) shows that in a global climate model in which the amount of $CO_2$ is doubled, the global mean surface temperature is underestimated if the amount of supercooled liquid is underestimated in mixed phase clouds. Korolev et al. (2017) provides an overview citing several examples including: 1. Large errors in the annual mean downwelling solar absorbed radiation at the Arctic surface (cloud albedo and optical thickness of the cloud will differ based on the liquid to ice ratio in the cloud, even for the same

total condensed water content), 2. Biases in the Arctic wintertime temperature inversion in many CMIP5 models, 3. Errors in the the development of precipitation and the lifetime of clouds (these depend on the ice phase specifically), and 4. Numerous





questions remaining about the cloud-aerosol indirect effects in mixed phase clouds (because a change in aerosol number is related to a change in liquid droplet concentrations, glaciation effects, riming effects, and so on). Emphasis is placed on a need for data in a large variety of surface, meteorological, and cloud conditions in the Arctic, including studies which examine the
inhomogeneity of clouds at small scales (i.e. spatial distribution of ice vs. liquid at scales as small as $10^{-1}$ m).

More recent weather prediction models have begun to include the distribution of the various phases of water within the mixed phase cloud (Forbes and Ahlgrimm, 2014). Vertical resolution has a large effect on mixed phase cloud model results. Barrett et al. (2017) use a vertical single-column model of mixed phase altocumulus clouds to perform resolution sensitivity studies. The low resolution model runs (500 m vertical resolution) reproduce only 12% of the mean supercooled liquid water content
which is present in the high resolution (50 m) run, and thus were unable to reproduce the supercooled liquid layer (~200 m thick) which is typical at the top of these clouds. The Barrett et al. (2017) model includes both ice and liquid microphysical process rates, and the difference in results between model runs is attributed to the dependence of microphysical process rates on vertical gradients of cloud properties. Models with simpler microphysics (no ice explicitly modeled) had less bias in total amount of supercooled water, but because they are not representing the clouds' vertical structure correctly (liquid layer atop
ice), they still underestimate the radiative impact of the supercooled liquid (Barrett et al., 2017). The Barrett et al. (2017) results are expected to apply to Arctic boundary layer clouds as well. Therefore, it is important to have both models and observations at high vertical resolution (tens, rather than hundreds of metres) for Arctic clouds.

Recent observational studies have been done on mixed phase clouds with supercooled liquid tops above ice-dominated cloud volumes (Morrison et al. (2012); Shupe et al. (2008), the same type of cloud modeled by Barrett et al. (2017)), and these are
referred to as "single-layer mixed-phase clouds" by, e.g. Mioche et al. (2015). Such single-layer mixed-phase stratiform clouds were observed by Boer et al. (2009) to occur with <5% frequency in spring, 5-8% frequency in summer, and 5-12% frequency in fall at Eureka, with most winter clouds being glaciated instead. These clouds are not the focus of the present paper.

Instead, we focus on a different type of mixed-phase cloud which contains layers of ice and liquid throughout the volume of the cloud. This type of mixed phase cloud has been less commonly studied, goes by a variety of names in the literature[1]
including "multi-layer clouds" (Mioche et al., 2015). Recent examples of multi-layer cloud studies are e.g. Verlinde et al. (2007, 2013), and Rambukkange et al. (2006), which look at layers <100 m thick, and McCullough et al. (2019), with layers <10 m thick.

Lidar is an excellent tool for studying mixed phase clouds as it produces vertical profiles of the atmosphere. With depolarization capability, the lidar can constrain the phase of the hydrometeors (liquid vs. ice), and thus represent the vertical distribution
of ice and water within the cloud as well (Schotland et al., 1971; Sassen, 2005; Bourdages et al., 2009; McCullough et al., 2017), the importance of which is described by Korolev et al. (2017). McCullough et al. (2019) showed measurements from the Canadian Network for the Detection of Atmospheric Change (CANDAC) Rayleigh-Mie-Raman lidar (CRL) at the Polar

---

[1]Cloud type nomenclature is not uniquely defined in the literature, in particular when layered and mixed-phase clouds are considered together. The clouds in our paper, with layers throughout the volume of a single cloud, are named "multi-layer" in the Mioche et al. (2015) nomenclature. However, this type of cloud is instead named "multilayered" in e.g. Vassel et al. (2019). In contrast, Vassel et al. (2019) uses "multilayer" (without a hyphen) for situations with two separate clouds (neither of which is necessarily a mixed-phase cloud) with a clear visible interstice in between. These "multilayer" clouds are the focus of Vassel et al. (2019) itself, and, e.g. Curry et al. (1988), but are not the topic of McCullough et al. (2019), nor of our present paper.





Environment Atmospheric Research Laboratory (PEARL; located at Eureka, Nunavut in the Canadian High Arctic (79.6° N, 85.6° W)). The laminations described in McCullough et al. (2019) are at least as thin as the detection limit of the lidar (7.5 m), which is an order of magnitude thinner than layers previously described in the literature. An example of such a laminated cloud

is shown in Fig. 1.

A cursory investigation in McCullough et al. (2019) indicated that laminated clouds occur throughout the year. Several questions arose: Are laminated clouds a significant feature at Eureka? How often do the laminated clouds occur? Are laminations ubiquitous, and therefore part of the background state of the local atmosphere, or are they infrequent events which perturb (or are the result of perturbations to) the background state? Therefore the need for a statistical investigation was indicated.

Questions relating to the makeup of the laminations themselves was investigated initially in McCullough et al. (2019) by looking at the full suite of CRL wavelengths, as well as depolarization and radiosonde measurements of relative humidity over ice and water, and windspeeds. Together, these measurements indicated that the laminations are associated with thermal/convective stability, precipitation, and that regions of high lidar backscatter are associated with low linear depolarization parameter, and vice versa. Therefore, it is likely that the laminations themselves are collections of either liquid droplets or

horizontally-oriented plate ice particles (high backscatter, low depolarization), and the regions between laminations are aerosols or randomly-oriented frozen particles (lower backscatter with higher depolarization). The laminations could also be related to the particle size and number density distributions. If we can quantify what kind of precipitation, or none, the laminated clouds are correlated with, this can further constrain our interpretation of the laminations themselves. We will learn more about what is the clouds are producing, and thereby learn more about the conditions and microphysical processes occuring within the cloud.

Therefore, an investigation linking laminations to surface weather was indicated.

The two goals for the current paper are:

**Investigation A:** Determine whether cloud laminations are a significant feature in the atmosphere at Eureka by determining the frequency, relative frequency, and monthly distribution of laminated clouds throughout the year.

**Investigation B:** Determine any correlation with meteorological conditions at Eureka, which will help improve our under-

standing of the makeup of, and processes within, laminated clouds.

To accomplish these goals, we quantitatively examine 3.5 years of measurements (January 2016 through June 2019). The datasets used are the CRL 532 nm range-scaled photocounts, and weather data from Environment and Climate Change Canada (ECCC) meteorological reports.

## 2   Method

### 2.1   Identification of laminated days using CRL

Measurements were made with CRL at 1 min × 7.5 m resolution up to 12+ km altitude. Plots of range-scaled 532 nm photocounts were produced for each 24 h UTC day from 0 to 5 km altitude at the measurement resolution (no further vertical integration). Shupe et al. (2011) have shown that the annual mean lowest cloud bottoms at Eureka are at 1.75 km altitude, while





mean highest cloud tops are at 4.5 km, exceeding 5 km only in August. Therefore, the altitude range used in the CRL study, up to 5 km, is deemed sufficient to encompass most clouds at Eureka. The CRL plots were examined by eye to determine whether laminated clouds could be detected at any time during that day. Days without laminations are further sub-classified. We desired to differentiate between days for which we can definitively show the presence or non-presence of laminations, and days for which CRL may not have been sensitive to laminations even if they were present above the lidar. This makes it possible to describe the relative occurrence frequency of laminated clouds.

The sky scene classification scheme is described below. There are 6 categories, with lower case roman numerals corresponding to the categories in the flow chart in Fig. 2. Corresponding example plots are given in Fig. 3. The categories are:

i. "No measurements". This category is for days on which CRL made no measurements at all, and therefore it is not possible to determine whether laminated clouds are present or not.

ii. "Laminated". This category identifies days, with any number or quality of measurements, which show a definitive detection of laminated clouds. The 24 h lidar scene counted as laminated if there were laminated clouds for any part of the day. The criteria used to identify laminated clouds were:

- The laminated region occurs in a cloud, and not only in a region of aerosol, dust, fog, etc.

- Defined edges of the cloud were visible; range-scaled-photocounts value was higher than surrounding non-cloud values by a factor of approximately 10 or higher.

- There were a minimum of three quasi-horizontal stripes, about equal in thickness, stacked one on top of the other, with maximum total extent of 75 metres from the lower edge of the first stripe to the upper edge of the third stripe.

- The laminated condition lasted for a minimum duration of 0.5 h.

Note that "Laminated" days are by definition cloudy. They are also considered to be "fully interpretable scenes" despite possibly having measurement gaps in time or altitude, because we can unambiguously determine that laminated clouds were present on these days.

iii. "Undetermined (obscured)". This category is for scenes in which low altitude clouds obscured at least 1 h of measurements for altitudes above those clouds. Laminated clouds were not detected during the measurement, but it is not possible to interpret the entire 24 h × 5 km sky scene, and therefore it is not possible to determine whether laminated clouds are present or not in the regions not represented in the lidar plot.

iv. "Undetermined (missing > 1 h data)". This category, like category iii, has no detections of laminated clouds. Because part of the 24 h scene is missing, it is not possible to determine whether laminated clouds present in the atmosphere on this day during the times that the lidar was not running.

v. "Non-laminated (cloudy)". No laminations were detected on these days. The entire 24 h × 5 km scene was visible, and no more than 1 h of measurements were missing nor obscured throughout the day. At least one cloud of duration 0.5 h or longer





was present. The day is considered "fully interpretable" because we can determine unambiguously that although there were clouds present, there were no laminated clouds.

vi. "Non-laminated (clear)". The entire 24 h × 5 km scene was visible, and no more than 1 h of measurements were missing nor obscured throughout the day. These days can include such features as aerosol layers, dust, etc., but no clouds, and may be more appropriately described as cloud-free. By definition, cloud-free days cannot contain laminated clouds. The day is considered "fully interpretable" because we can determine unambiguously that there were no laminated clouds present.

As indicated in the list, three categories are considered to be fully interpretable: i. Laminated (which is by definition cloudy), v. Non-laminated (cloudy), and vi. Non-laminated (clear). These categories all unambiguously describe the (non-)laminated status of the atmosphere on that day. The remaining three categories are not fully interpretable, and therefore lead to indeterminate results: iii. Undetermined (obscured), v. Undetermined (missing >1 h data), and i. No measurement. While we did not detect laminations in these indeterminate scenes, it is possible that there were laminations present those days. For this reason, the lamination statistics in this paper indicate minimum occurrence frequency.

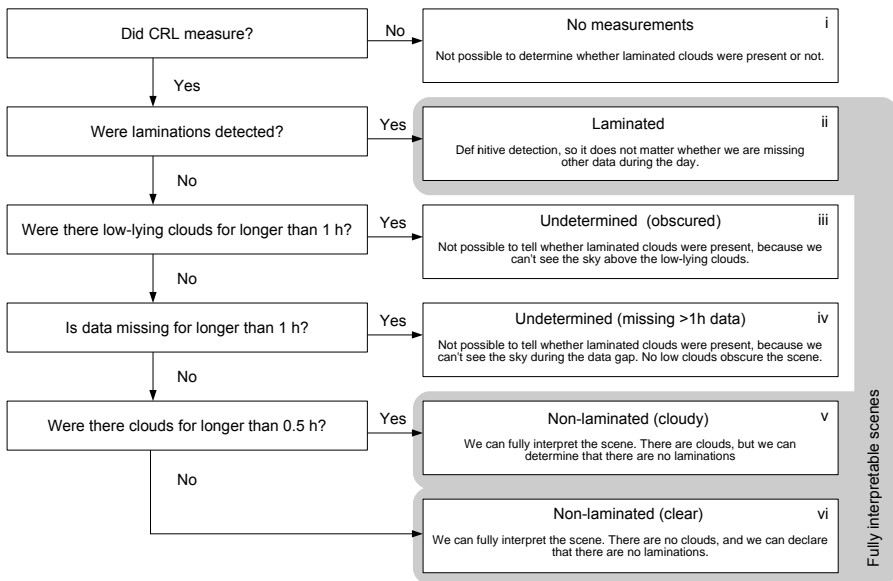

**Figure 2.** Flow chart describing the classification of CRL sky scenes into three interpretable categories: ii. Laminated, v. Non-laminated (cloudy), and vi. Non-laminated (clear), and several undetermined categories: i. No measurements, iii. Undetermined (obscured), and iv. Undetermined (missing > 1 h data).

## 2.2 Weather from ECCC

Publicly available meteorological reports from Environment and Climate Change Canada (ECCC, formerly Environment Canada (EC)) were used for context such that we might investigate the conditions at the ground which are consistent with laminated clouds as identified by CRL.

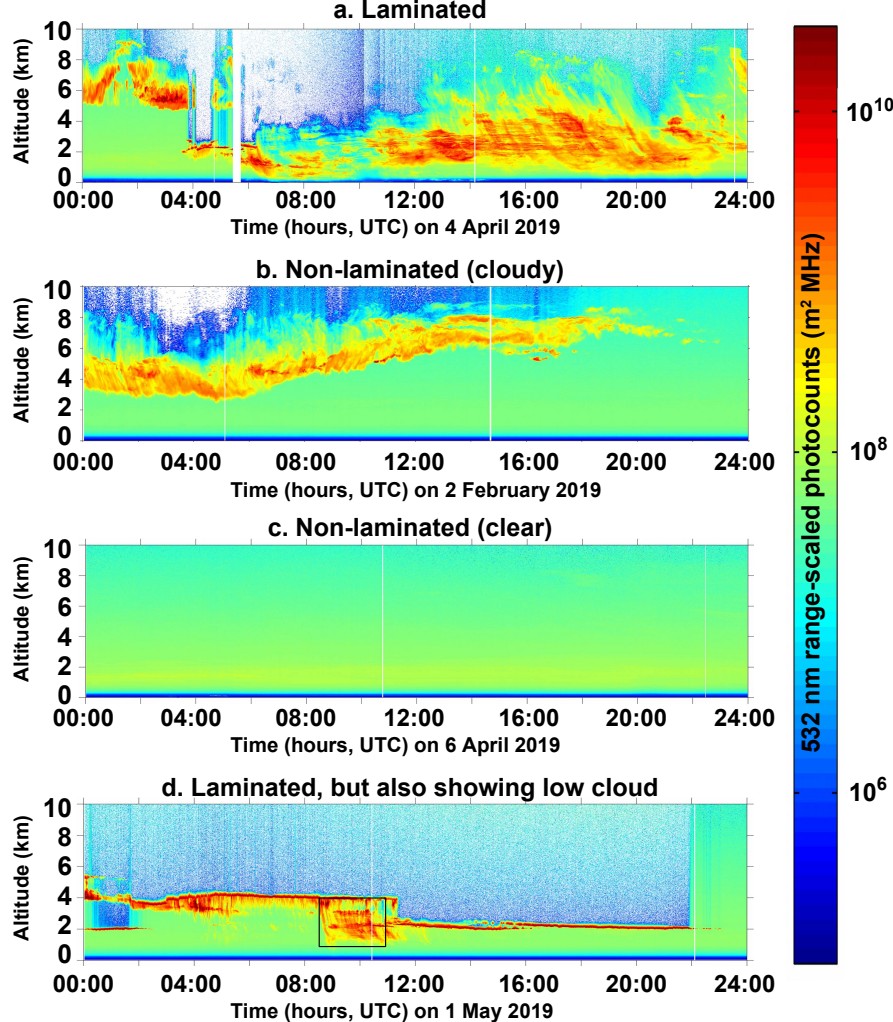

**Figure 3.** 532 nm range-scaled photocount plots associated with sky scene categories in Fig. 2. 1 min x 7.5 m resolution, 0 - 10 km shown for context. Panel a: "Laminated". Laminations are visible 07:00 - 24:00 UTC. The white vertical bar is a gap in the measurements. Even if this gap had exceeded 1 h duration, it would not impact our classification because laminations are definitively seen. On a non-laminated day (e.g. Panels b,c), a gap > 1 h would cause reclassification to "Undetermined (missing > 1 h data)". Panel b: "Non-laminated (cloudy)". There are clouds below 5 km which do not obscure other altitudes in the scene up to 5 km, and none of the clouds are laminated. Panel c: "Non-laminated (clear)". There are no clouds, and therefore no laminated clouds. There are aerosol signatures visible below 1 km all day, and at 18 UTC at 8 km do not impact our classification. Panel d: "Laminated". Laminated cloud is visible at 10:00 UTC (The 3 layers visible at this scale are not the laminations - the laminations are the finer features visible in the expanded excerpt in Fig. 1.) There is also low-lying cloud identified 12:00 UTC - 22:00 UTC on this day, but because laminations have been detected, the day counts as "Laminated". If there were no visible laminations present, we would classify the day as "Undetermined (obscured)" instead. Black box expanded as Fig. 1.





**Table 1.** ECCC reported weather conditions were grouped together for comparison in this paper.

| Weather category | ECCC weather conditions included |
| --- | --- |
| All Precipitation | snow, moderate snow, snow showers, snow grains, snow pellets, rain, rain showers, drizzle, ice pellets |
| All Snow | snow, moderate snow, snow showers, snow grains, snow pellets, ice pellets, blowing snow |
| Snow | snow, moderate snow, snow showers, snow pellets |
| Blowing Snow | blowing snow |
| Fog | fog |
| Rain | rain, rain showers, drizzle, freezing drizzle |
| Ice Crystals | ice crystals |
| Freezing Fog | freezing fog |
| Excluded from manuscript | clear or mainly clear, cloudy or mostly cloudy, dust |

The ECCC weather data are recorded by human weather observers at Eureka. The data are generally reported hourly, with between 22 and 24 observations being made per day. For each day of ECCC meteorological data, we identified a positive detection for any type of weather which was reported at any time during that day, and non-detection for any weather which did

not occur that day.

For analysis in this paper, we have grouped the ECCC standard weather observations into categories, as in Table 1. The first two categories in the table, "All Precipitation" and "All Snow" are combinations of the categories lower in the table. The subsequent categories are grouped logically by type of weather and are all independent of one another. We have excluded three types of reported weather from consideration: "Dust", which only appeared on one measured day in 2017, and "Clear or mainly

clear", and "Cloudy or mostly cloudy", descriptors which are only used in the absence of any other type of weather occurring that hour. For example, an hour with "snow" will not be listed also as "cloudy" in the ECCC weather data, despite clouds being required in order for the snowing condition to be reported. "Ice Pellets" (2 instances) and "Snow Grains" (4 instances) have been included in the "All Snow" category, but excluded from consideration otherwise.

### 2.3    Correlation of CRL laminations with ECCC weather

Pearson product-moment correlation coefficients, r, and the associated significance values, p, were calculated for the three CRL fully interpretable categories (Laminated, Non-laminated cloudy, and Non-laminated clear), and our weather categories listed in Table 1.

The Pearson's r is expressed as the covariance of two random variables divided by the product of the standard deviation of each variable $X$ and $Y$, as in: $r = cov(X, Y)/\sigma_X * \sigma_Y$.

The interpretation of r is discussed at length in Cohen (1988) as being useful only in a relative sense; context is required in order to state whether a particular value of r is "small" or "large". Thus, all r values reported here are most useful when compared with one another. The convention suggested by Cohen (1988) is what we will follow for descriptors in this paper:





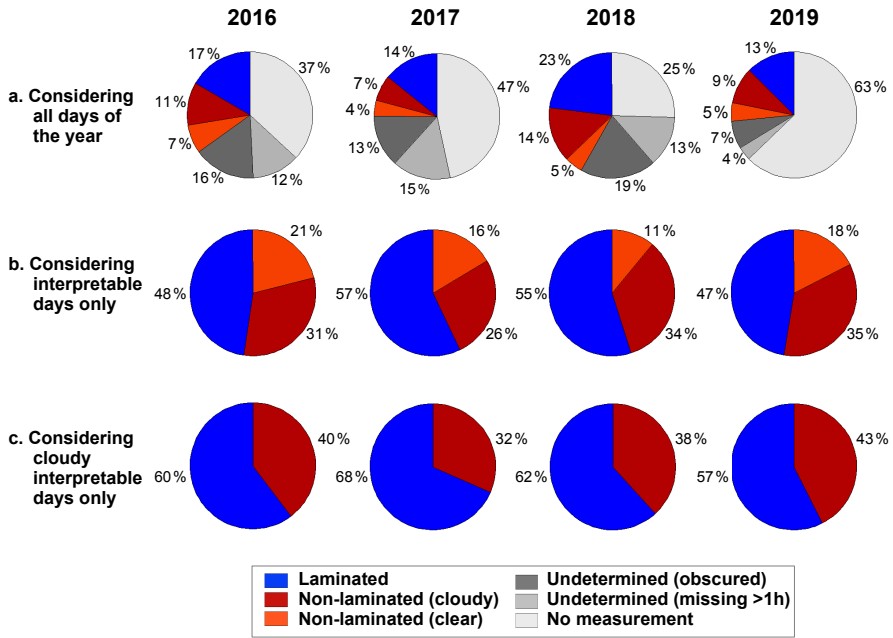

**Figure 4.** Frequency of laminated clouds compared to other sky scene classifications from CRL.

$r = 0.1$ is *small*, $r = 0.3$ is *medium*, and $r = 0.5$ is *large*. We are working with a 95% ($2\sigma$) confidence interval, so we take r values having p $\leq$ 0.05 to be significant.

We use r to express how the behaviour of the local weather conditions from ECCC varies with the cloud determination made
5  by the lidar.

In our study, daily results were compared. A certain type of weather may be noted for a particular day, but may not have occurred at the same time as the laminated clouds, if neither lasted the entire day. Consequently, there will be some amount of noise expected in the final result, and the correlations are thus intended as minimum correlation amplitudes. We seek the combinations of lidar cloud morphology and reported weather condition which have higher correlations than other combinations
10  have. Tables of correlation coefficient r were colour-coded for ease of interpretation, in the style of a corrgram, or correlogram.

Comparisons are also expressed in this paper as relative percentages, showing how much more likely snow is to occur on a day with laminated clouds than it is on a day with non-laminated clouds, for example, as this is an intuitive way to understand the results.





**Table 2.** Occurance of laminated clouds for each year, a. Considering all days of the year (see note below table about 2019*), b. Considering only the interpretable days, and c. Considering only the interpretable cloudy days.

| | 2016 | 2017 | 2018 | 2019 | 2019* | All years |
|---|---|---|---|---|---|---|
| a. Considering all days of the year | N = 366 | N = 365 | N = 365 | N = 365 | N = 181 | N = 1277 |
| Laminated | 61 (17 %) | 52 (14 %) | 84 (23 %) | 46 (13 %) | 46 (25%) | 243 (19.0 %) |
| Non-laminated (cloudy) | 40 (11 %) | 24 (7 %) | 52 (14 %) | 34 (9 %) | 34 (19 %) | 150 (12 %) |
| Non-laminated (clear) | 24 (7 %) | 15 (4 %) | 17 (5 %) | 17 (5 %) | 17 (9 %) | 76 (6 %) |
| Undetermined (obscured by low cloud) | 58 (16 %) | 49 (13 %) | 71 (19 %) | 26 (7 %) | 26 (14 %) | 204 (16 %) |
| Undetermined (missing > 1 h data) | 45 (12 %) | 55 (15 %) | 48 (13 %) | 13 (4 %) | 13 (7 %) | 161 (13 %) |
| No measurement | 135 (37 %) | 170 (47 %) | 93 (26 %) | 229 (63 %) | 45 (25 %) | 443 (35 %) |
| b. Considering only interpretable days | N = 128 | N = 91 | N = 153 | N = 97 | N = 97 | N = 469 |
| Laminated | 61 (48 %) | 52 (57 %) | 84 (55 %) | 46 (47 %) | 46 (47 %) | 243 (52 %) |
| Non-laminated cloudy | 40 (31 %) | 24 (26 %) | 52 (34 %) | 34 (35 %) | 34 (35 %) | 150 (32 %) |
| Non-laminated clear | 24 (21 %) | 15 (17 %) | 17 (11 %) | 17 (18 %) | 17 (18 %) | 76 (16 %) |
| c. Considering only cloudy interpretable days | N = 101 | N = 76 | N = 136 | N = 80 | N = 80 | N = 393 |
| Laminated | 61 (60 %) | 52 (68 %) | 84 (62 %) | 46 (57%) | 46 (57%) | 243 (62 %) |
| Non-laminated cloudy | 40 (40 %) | 24 (32 %) | 52 (38 %) | 34 (43 %) | 34 (43 %) | 150 (38 %) |

Note: For 2019, only measurements 1 January to 30 June are included in the analysis as these were available at the time of writing the manuscript. Therefore, about half of the value noted for 2019 as "No measurement" are dates in the future, which skews the results. The "2019*" column truncates at 30 June 2019 and recalculates the percentages based on a year which is 181 days long. This is equivalent to assuming that August through December has the same distribution of measurements and sky conditions as the first half of the year. The "All Years" column includes 2016, 2017, 2018, and 2019*, covering 3.5 years.

# 3 Results and Discussion for Investigation A: Frequency and distribution of laminated clouds throughout the year

## 3.1 Yearly frequency results

The sky scene classification of each day of the year for 2016 through 2019 according to the 6 categories in Sec. 2.1 is represented by the pie charts in Fig. 4a and Table 2a. A total of 834 days were measured by CRL from 2016 through June 2019 (present date at writing of this manuscript). There are definitive laminated cloud detections on 243 of these days, which averages to 70 days per year. This includes 61, 52, and 84 days per full year (2016, 2017, 2018), and 46 detections in the 2019 half-year. Excluding the half-year (in case a non-uniform distribution of laminated clouds throughout the year is relevant), the average is 66 days per year. Therefore, a minimum of approximately 20 % of the days of the year have laminated clouds. There may be additional laminated clouds occurring in any of the undetermined categories (grey colours in Fig. 4), which occupy two thirds to three quarters of each year.





**Table 3.** Monthly distribution of measurements. Totals are the sum for all measured days in each named month for the 3.5 year study period. Total number of interpretable days is the sum of the first three rows of the table. Total number of cloudy interpretable days is the sum of the first two rows of the table.

|  | Jan | Feb | Mar | Apr | May | Jun | Jul | Aug | Sep | Oct | Nov | Dec |
|---|---|---|---|---|---|---|---|---|---|---|---|---|
| Laminated | 1 | 9 | 52 | 44 | 36 | 24 | 10 | 8 | 14 | 20 | 19 | 7 |
| Non-laminated (cloudy) | 4 | 22 | 29 | 22 | 17 | 11 | 12 | 5 | 3 | 5 | 15 | 5 |
| Non-laminated (clear) | 2 | 7 | 13 | 12 | 18 | 3 | 4 | 5 | 1 | 3 | 6 | 2 |
| Laminated % of interpretable days | 14.3 | 23.7 | 54.8 | 56.4 | 50.7 | 63.2 | 38.4 | 44.4 | 77.8 | 71.4 | 47.5 | 50.0 |
| Laminated % of cloudy interpretable days | 20.0 | 29.0 | 63.8 | 66.7 | 67.9 | 68.6 | 45.5 | 61.5 | 82.3 | 80.0 | 55.9 | 58.3 |

In Table 2b and Fig. 4b, only the fully interpretable days are considered. The sky may be cloudy or clear, but the full 24 h x 5 km scene is visible and interpretable for each of these days. Laminated clouds are detected on 47 % to 57 % of the the measurement days each year for which we have appropriate data to be able to detect them if they are present.

Finally, a subset of the interpretable days is examined: Only those which are cloudy are included in Table 2c and Fig. 4c. Laminations are detected on between 57 % and 68 % of the cloudy days for which we have appropriate data to be able to detect them if they are present.

## 3.2 Monthly frequency results

The sky scene classification is plotted monthly in Fig. 5, and the monthly values for all years combined are given in Table 3, to explore the distribution of laminated clouds throughout the year. Laminated clouds can and do occur at all times of year. Indeed, February 2016 is the only measured month in 3.5 years that shows no laminations, and that particular February there were only 4 interpretable days measured.

Monthly detection rates of laminated clouds are seen in Fig. 5. We can see the number of laminated cloud days which were definitively detected in each month. The true occurence frequency of laminated clouds is therefore likely to be higher as measurements were not made on every day of each month, and not all measured days were interpretable - therefore this is a minimum number understood to exist in the atmosphere at Eureka. March, April, and May have definitive detections of laminated clouds on 25 % to >50 % of all days of each month. October and November show similar detection rates. June, July, August and September, as well as any Decembers, Januaries, and Februaries which have more than a couple of measurement days, show somewhat lower numbers of detections, generally 5 % to 20 % of all days of the month.

Removing the effect of the number of measurement days, we find that relative detection rates for laminated clouds vary between 14 % and 78 % of all interpretable days, and between 20 % and 82 % of all cloudy interpretable days, depending on the month.

**Figure 5.** Monthly bar charts as of 30 June 2019. Measurements are ongoing. Colours correspond with those in Fig. 4. Percentages are with respect to all (28 to 31) days in each month. Only the interpretable days are shown in upper panels, for clarity.





20     Considering all interpretable measured days, January and February have the fewest days with laminated clouds, at 14.0 % and 23.0 % respectively. July and August have about double as many detections, at about 40 %. November, December, and May show yet slightly higher rates, exhibiting laminated clouds on about 50 % of all interpretable measured days. March and April are the first months for which any measured interpretable day is more likely than not to show laminated clouds; they show about 55 % each. In June, the frequency increases by another 10 % to 63.2 %. September and October show the highest values of the year, between 70 % and 80 % occurence frequency; during this time of year there are laminated clouds on about three-quarters of all fully interpretable measured days.

    The values when considering only cloudy interpretable days are uniformly larger, by a factor of 1.1 to 1.4, than the values
5  when clear interpretable days are included. Again, January and February have the lowest detection rates for laminated clouds, representing approximately 20 % and 30 % of all cloudy interpretable days, respectively and July has the next higher value at 45 %. November and December both have occurance frequencies greater than half, above 55 % each. March, April, May, June, and August all have values between 60 % and 70 %. September and October have the highest relative frequencies when considering only cloudy interpretable days, at over 80 % each.

## 10   4   Results for Investigation B: Correlation of laminations with weather

### 4.1   CRL cloud lamination - ECCC weather correlation results

Considering only the three fully interpretable categories, laminated, non-laminated cloudy, and non-laminated clear, there were a total of 128, 91, 153, and 97 measurement days for 2016, 2017, 2018 and 2019 respectively, to use in the comparisons with the 8 categories of ECCC weather data.

15     Pearson's r correlation values were calculated for each of the combinations discussed in Sec. 2.3. The correlation results are provided in Fig. 6. Correlation scores can vary between $r = -1$ (complete anti-correlation) and $r = 1$ (complete positive correlation). In Fig. 6, stronger correlations are highlighted with darker shades of red or blue and significatant values of r are in bold.

    In all years, the highest correlation values were between laminated clouds and snow. Three years, 2016, 2017, and 2018,
5  yielded a large correlation, with r values of 0.63, 0.67, and 0.56 and 2019 yielded a medium correlation of 0.42, however, we have yet to include end of year measurements in this total. Snow was significantly negatively correlated with non-laminated cloudy and clear sky scenes in all years, with medium negative correlation values.

    Blowing snow was also positively, but weakly, correlated with laminated clouds, with r values of 0.23 for 2017 and 0.26 for 2018, the only two values which were significant. 2016 and 2019 show weakly positive but, non-significant correlations.
10  Blowing snow is has significant small negative correlations with non-laminated cloudy days in 2018, and with non-laminated clear days in 2019.

    The combined All Snow category produces correlations which are much weaker (in 2016), slightly stronger (in 2017), and slightly weaker (in 2018 and 2019) than the correlation with snow alone. All Snow is likewise negatively correlated with non-laminated cloudy and clear days, with small to medium correlations.





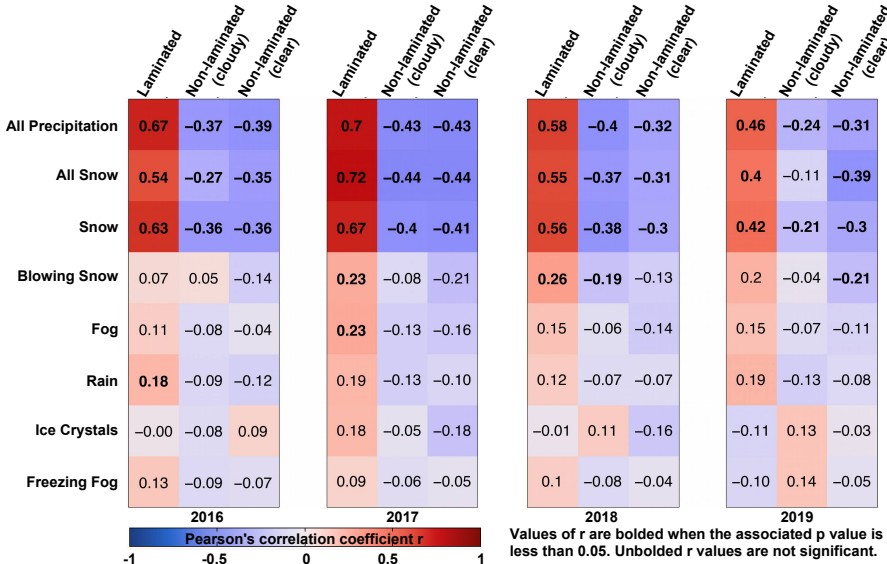

**Figure 6.** Correlation between Laminated, Non-laminated (cloudy), and Non-laminated (clear) skies from CRL and weather categories from ECCC. Pearson's r correlation coefficient is plotted in colour with each value identified. Red indicates a strong positive correlation, white indicates no correlation, and blue indicates a strong negative correlation. Values in bold are significant at $2\sigma$.

The combined category with all precipitation (snow and rain together) retains a significant positive, and medium-to-strong, correlation of 0.67, 0.7, 0.58 and 0.46 for each year. This combined value which is no doubt dominated by the high number of snow days as compared to rain days, is nonetheless higher in each case than the correlation value for snow alone.

Rain displayed correlations of about 0.2 for each year, but this correlation was only significant for 2016; this insignificant correlation is likely due to the low number of rainy days at Eureka which do not also display low altitude clouds.

Ice crystals and freezing fog displayed no significant correlations for any year. Fog had a significant positive correlation with laminated cloudy days in 2017.

The correlation plots of Fig. 6 provide clear evidence of the relationship between snow and laminated versus non-laminated clouds. A more intuitive way to explore this result is to examine the relative occurance frequencies of each combination. Table 5 in Appendix A gives the number and percent of laminated, non-laminated cloudy, and non-laminated clear days on which each type of weather occurred for at least 1 h. Combined all-year values are given in Figure 7 and Table 4 for combinations which have significant Pearson's r correlation values in at least one year.

Precipitating snow occurs about 5 times more often on days with laminated clouds compared to days with non-laminated clouds. Rain occurs 7 times as often. Blowing snow and fog each occur about twice as often on laminated cloud days. The details are as follows.

All Precipitation, which includes rain and precipitating snow, is 4.7 times more likely to occur on laminated cloudy days than it is on non-laminated cloudy days (71 % and 15 % occurence frequency, respectively). Precipitating snow, with 183 measured





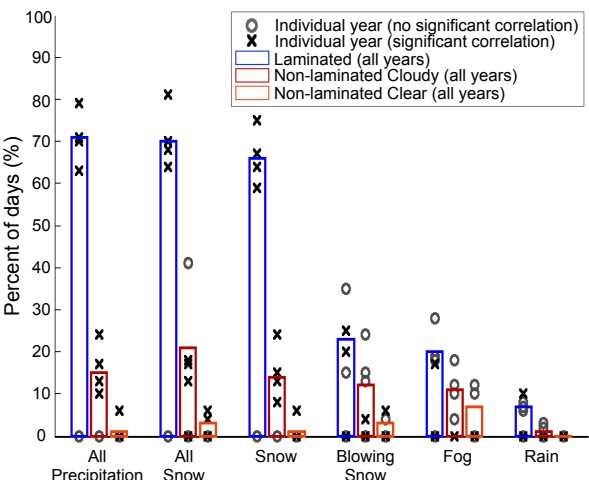

**Figure 7.** Percent of days for which each type of weather was reported for at least 1 h, for each of the three fully interpretable sky scene categories. Bars show results for all years 2016 - 2019 combined, as in Table 4. Points are data from individual years, as in Table 5, coloured according to correlation significance in Fig. 6. Black points show significant correlation between the sky scene classification and the weather for that year, and grey points show no significant correlation.

**Table 4.** Number of all fully interpretable days for which each type of weather was reported for at least 1h, with percentage in brackets. All years 2016 - 2019 combined. Only weather with at least one significant correlation (Fig. 6) are included. See Table 5 for individual years and remaining weather categories.

|  | Laminated | Non-laminated Cloudy | Non-laminated Clear |
|---|---|---|---|
|  | % | % | % |
| Number of interpretable days | N = 243 | N = 158 | N = 76 |
| All Precipitation | 173 (71 %) | 24 (15 %) | 1 (1 %) |
| All Snow | 170 (70 %) | 33 (21 %) | 2 (3 %) |
| Snow | 161 (66 %) | 22 (14 %) | 1 (1 %) |
| Blowing Snow | 55 (23 %) | 19 (12 %) | 2 (3 %) |
| Fog | 49 (20 %) | 17 (11 %) | 5 (7 %) |
| Rain | 18 (7 %) | 2 (1 %) | 0 (0 %) |

days, dominates this effect, and is itself 4.7 times more likely to occur on laminated cloudy days (66 % occurence frequency) than it is on non-laminated cloudy days (14 % occurence frequency).

15    The contribution of rain to the All Precipitation category is small: only 20 measured days. This small number of rain days is also responsible for the insignificant correlation r values in Fig. 6. Nevertheless, the relative occurence frequency values for





rain remain interesting and would benefit from more study once further years' data are available. Rain occurs 7 times more often during laminated cloudy days (7 %) than it does during non-laminated cloudy days (1 %; only 2 days over the entire 3.5 year study period).

Considering the All Snow category (precipitating and blowing), laminated days (70 % occurence frequency) are 3.3 times more likely to display any type of snow than non-laminated cloudy days are (21 % occurence frequency). During the study period, there were similar numbers of non-laminated cloudy days having blowing snow (19 days) and precipitating snow (22 days), but the totals are vastly different for laminated days: 55 blowing snow days, compared to 161 precipitating snow days. Therefore, precipitating snow is primarily responsible for the correlations in the All Snow category as well. Blowing snow occurs twice as often on laminated cloudy days (23 %) as it does on non-laminated cloudy days (12 %), which is an effect less than half the size of that for precipitating snow.

Fog showed significant correlation in 2017, with an r value larger than that found for the significant 2016 correlation for rain. However, while rain showed a 7-fold difference in occurance frequency between laminated and non-laminated cloudy days, fog shows only a 2-fold difference (20 % versus 11 %), which is a result more akin to that of blowing snow.

# 5   Discussion

## 5.1   Discussion for Investigation A

Investigation A seeks to quantify the laminated clouds in the Arctic atmosphere: How often do the laminated clouds occur, and are they therefore likely to have an important or significant effect on the overall state of the atmosphere at Eureka?

The finding that laminated clouds occur on average a minimum of 70 days per year, which accounts one-fifth of the year, indicates that laminated clouds do occur with sufficient frequency that they are relevant to our overall understanding of the atmospheric conditions. This minimum occurrence frequency is found despite a relatively high number of unmeasured days per year, and an additional number of days which are not fully interpretable by the criteria laid out in Section 2.1. It is almost a certainty that the true occurence frequency of laminated clouds at Eureka is much higher - and thus that they have an even stronger impact on the atmosphere than might be estimated using the value of 70 days per year. These laminated clouds are not stochastic, infrequent features - rather, they are part of the general situation.

More importantly for cloud-specific studies, approximately 50 % to 60 % of all fully interpretable days, cloudy and clear, exhibit laminated clouds. Therefore, they are present on more than half of all measured days. Further, any fully interpretable cloudy day is much more likely than not to exhibit laminated clouds, with a relative occurence frequency of approximately 60 % to 70 %. This is a finding that cannot be ignored when probing the microphysical processes and the dynamics within clouds between 0 and 5 km at Eureka.

Regarding the distribution of laminated clouds throughout the year, it is expected that the results are influenced by the small number of measurements made during some months.

March, April, and May have the highest number of measured days, principally because of the ACE/OSIRIS Arctic Validation Campaign which occurs annually during polar sunrise (see for example Kerzenmacher et al. (2005), Adams et al. (2012), Griffin





et al. (2017) and references therein), and measurements tend to be easily continued following the campaign each year. Further, these months tend to have very few closures longer than 1 h (because there were few days with high winds, rain, etc., which could damage the lidar if it were to attempt measurements), so we can be confident that CRL is getting an accurate picture of the sky. In these three months, laminated clouds are measured on about 50 % of all interpretable days, and on more than 50 % of all cloudy interpretable days. This is a clear indication that laminations in Arctic mixed-phase clouds at Eureka are an

integral component of the atmosphere during the spring season.

Summer months, June, July, August, and September, have frequent fog and cloud at low altitudes which obscures the sky scene above, as determined by this study. (Shupe et al. (2011) notes low level clouds on average 40 % of the time, and 20 % of the time in summer). If there are hidden laminated clouds above the optically thick clouds, we may be severely under-counting lamination events especially during summer. Even so, of the fully-interpretable days in these months, there is a better than 50

% chance of seeing laminated clouds. Therefore, we expect that laminated clouds may form an even larger component of the atmosphere at Eureka during summer than the values measured here are capable of showing.

October and November generally display more laminated than non-laminated clouds on interpretable days, and display laminated clouds on at least 30 % of all days of the month (measured or not).

December, January, and February each year have zero to few measured days for CRL because of funding and operational

constraints of the Polar Environment Atmospheric Research Laboratory. Therefore, although the "percent of days with laminations" in Fig. 5 is a small number, it is based on so few measurements that it is not yet a particularly significant finding. Better estimates of the mothly laminated cloud frequency would be possible with increased funding to operate CRL during the December to February period each year.

Cloud laminations seem to be a ubiquitous feature of the lower troposphere at Eureka, and the 3.5 studied years are very

similar to each other, so we infer that 3.5 years of measurements is of ample duration to act as a case study when making more detailed investigations as to the nature of the laminations.

## 5.2 Discussion for Investigation B

Prior to the correlation analysis, we were unsure whether snow, blowing snow, or both together, would be correlated with laminated cloudy days. The All Snow category was included in this study due to a practical aspect of weather observation

at Eureka station, which may allow the Snow and Blowing Snow categories to become somewhat conflated: The weather conditions (snow, blowing snow, ice crystals etc.) are reported using observations made at Eureka Weather Station, at 10 m above sea level. The windspeeds are recorded at the airport runway, which is 1.8 km to the Northeast, and 73 m higher in elevation, than the weather station's instrument compound. Due to ECCC's reporting criteria, blowing snow is not able to be reported when the windspeed is less than a minimum value. There is the possibility of a confounding variable in the

blowing snow category arising from local wind speed. Note that CRL lidar is located 180 m to the Northwest of the weather station's instrument compound. McCullough et al. (2019) established no relationship between laminated clouds and wind speed. Changes in the reporting methods may account for some of the relative increase in the number of blowing snow cases year by year. Therefore, the All Snow category was included in the study, to group these two cases together, in addition to





studying each separately. Blowing snow is often reported on days which also report snow, so sometimes the All Snow category will be affected more than other times.

From the results in Section 4, we see that it is truly the precipitating snow, rather than the blowing snow, which is correlated with laminated clouds. This is also consistent with the finding from McCullough et al. (2019) that laminations occur in a variety of wind conditions, high winds being required for blowing snow.

McCullough et al. (2019) made a preliminary report of several cases of coincident laminated clouds and rain, however the more quantitative analysis coving the full 3.5 year period indicates only a small positive correlation with rain. Rain has more than one formation mechanism in clouds. Raindrops can form directly from water vapour (condensing droplets, conceivably from convective clouds with vertical mixing, in which we would expect no persistent laminations) or can form indirectly, from melting snowflakes (possibly from laminated clouds, which we have shown to be strongly correlated with precipitating snow).

This confounding variable may contribute to the weak correlation of rain with cloud laminations. Perhaps only certain kinds of raindrops which have formed via specific processes are correlated with laminated clouds. Adding more years' data into this study might reveal more significant small correlations of rain with laminated clouds, or rule them out. This same discussion can be applied to the All Precipitation category, already discussed - it makes sense that if only some kinds of rain are correlated with laminations, that the All Precipitation category will be less well correlated with laminations than the Snow category is.

Certainly, the stronger correlation with snow is enough to explain many instances of cloud laminations in cold conditions.

Snow and rain are formed in clouds, so it is not surprising that these forms of precipitation have significant negative correlation with clear/cloud-free sky conditions. More interestingly, snow is also significantly negatively correlated with Non-laminated cloudy conditions. Precipitating snow is 5 times more likely to occur on a cloudy day if the clouds exhibit laminations. Rain is 7 times more likely if the clouds are laminated, compared to if they are not. This points to formation mechanisms

of snow, and perhaps rain as well, in relation to cloud appearance in high resolution lidar data. This contrast in correlation values between laminated and non-laminated clouds, in otherwise identical sky conditions (fully sensed up to 5 km for 24 h, with clouds visible), is a key result of this paper: The precipitating snow is correlated with laminated clouds, and is negatively correlated with non-laminated clouds.

### 5.3   Horizontal variability of mixed phase clouds

One interesting possibility to follow up on is the horizontal variability within mixed phase clouds. It is possible that the laminations, which arises as a term describing visual features of lidar image plots, may in fact arise from a geophysical feature which is in fact patches or clusters of liquid droplets or ice crystals, rather than a contiguous layer. As the feature passes by the lidar during the 1 min integration time, this could have the effect of spreading out the horizontal extent of the feature from the perspective of the lidar. A patch of hydrometeors which is present for part of the 1 min time bin is displayed as present

during that entire 1 min. Therefore, discrete horizontal features may appear connected in the lidar data. For CRL, any gaps smaller than approximately 1 m in horizontal extent will not be resolvable at 5 km altitude from the lidar, due to the divergence of the laser beam, even for a single laser shot. When integrating for 1 min, at the 2 to 15 m/s windspeeds found at 5 km in McCullough et al. (2019), gaps smaller than approximately 120 to 900 m become undetectable.



The small scale horizontal variability of clouds has been shown by García et al. (2012) and more recently by Ruiz-Donoso et al. (2019) to be an important consideration when calculating the total upwelling and downwelling radiation budgets. Therefore, a 3-D approach will eventually be necessary to understand the complete impact of laminated clouds on the radiation budget.

The Wegener-Bergeron-Findeisen (WBF) process, whereby ice crystals grow using vapour supplied from liquid droplets in the same cloud, is one example of a process which can result in patchy features (Tan and Storelvmo, 2016). The entire cloud does not glaciate at a uniform rate. Instead, clusters of ice particles and clusters of liquid droplets are formed. The horizontal size of these clusters can be as large as a few kilometers (Korolev et al., 2003; Field et al., 2004) or, in the Arctic, as small as 10 m (Chylek and Borel, 2004). Global circulation models which inappropriately assign a homogenous mixture of ice and liquid throughout an entire cloud volume can experience a WBF process which is too efficient, and lasts for timescales six orders of magnitude shorter than those observed (Tan and Storelvmo, 2016). With our vertically-resolved measurements, we access information about one dimensional size of this patchy process, but, as described above, CRL is insensitive to horizontal inhomogeneity smaller than ~100 - 900 m at 5 km altitude.

Combining the lidar with other instruments as was done in Ruiz-Donoso et al. (2019) would be one way to access 3-D cloud information. Combining CRL measurements with windspeed information can provide some upper limits on the horizontal spacing between laminated cloudy regions, but the zenith-pointing lidar is effectively a 1-D tool. Therefore in this paper we have focused on the vertical information.

Despite the scientific motivations for considering the possibility of patches versus laminations, the results in this paper are not changed. If it is the case that the laminations are from a collection of discrete patches, the patches still need to be confined to relatively contiguous, closely stacked (<10 m thick), altitude levels for timescales of hours. Thus, we may be discussing laminations of pocket-containing and non-pocket-containing layers, rather than laminations of homogenous regions, but the laminations persist. Therefore, while investigating these features at higher time resolution is interesting, the possibility that some patchy features are being interpreted as laminations does not undermine the results found in the current paper.

## 5.4 Future work

There is a reasonable way forward for improving the completeness of the study for Investigations A and B, should better statistics be desired.

Classifying each day is labour intensive. Adding a few months' worth of campaign data from 2009 through 2013 would not suffice to build up a climatology, so that is not a goal of this paper. We will nonetheless continue measurements to see whether the results here are part of a trend, and whether there is something else we did not account for in the data or interpretation, for example the total snowfall for the year.

In Section 2.1, the Undetermined categories place a tight restriction on data quality. The days for which there was only an hour or two of low-lying cloud, and no visible laminations, are all combined together in the Undetermined (obscured) category with days containing 24 h of low-lying cloud. Likewise, days missing an hour or two of data are in the same category as those missing 23 h, the Undetermined (missing >1 h data) category. For many of these days, it may be reasonable to guess, or





interpolate, the sky conditions during the missing hours. When the sky has been clear for a number of hours before and after the missing data, and the winds have not changed, it is likely that the sky during the missing hours was clear, too.

In this study, as long as the laminations and a particular kind of surface weather occur on the same day, it was not necessary that the laminations occur precisely at the same time as that weather in order for the coincidence to be counted. Future studies of a subset of data could reduce the uncertainty in the results by using the hourly resolution of the ECCC data, and redoing the CRL sky scene classification hourly to match. Nevertheless, the results are strong enough using the daily approach that this has not yet been necessary in order to conclude that snow, and no other meteorological condition, is the weather most strongly correlated with laminated clouds.

Further, there is a Millimeter Wavelength Cloud Radar (MMCR, Moran et al. (1998)) colocated with CRL. While it cannot detect the laminations (insufficient resolution), it can determine whether a cloud was present that could have hosted laminations. If the MMCR shows totally clear sky, then laminations are highly unlikely. Likewise, if it shows only clouds with a morphology inconsistent with laminations. Therefore, we can likely improve the statistics in this paper, and reduce the number of days falling into one to two of the "Undetermined" categories, if we loosen the data quality criteria or include insight from colocated instruments.

Given the strong results from the existing database studied in this paper (laminated clouds being ubiquitous throughout the year at Eureka; strong correlation between precipitating snow and laminated clouds), we are interested in following up on the precipitation-formation processes within these clouds. To that end, we will be selecting several representative case studies on snowing laminated cloud days to examine in detail with such measurements as lidar depolarization, radar, radiosonde, and other colocated instruments at Eureka. We are also interested to seek out results from other polar lidars to see whether the same laminated features are present at those locations as well, and how their appearance and correlations with surface weather compare to those found here at Eureka.

## 6 Conclusions

Key conclusions from Investigation A: Frequency of laminations:Cloud laminations are found to be a significant feature in the CRL measurements. Laminations occur often at Eureka. There are laminated clouds on 52 % of all interpretable days, and on 62 % of interpretable cloudy days. The relative frequency of laminated clouds to non-laminated clouds is generally consistent year to year, within the 3.5 year study carried out. There is a minimum average of 70 laminated cloud days detected per year, with no studied full year having fewer than 52 detections. and it is probable that the true occurence frequency of laminated clouds at Eureka is much higher.

Key conclusions from Investigation B: Correlation of laminations with weather: Precipitating snow is strongly correlated with laminated clouds, and is anti-correlated with non-laminated clouds. Precipitating snow is 5 times more likely to occur on a day with laminated clouds as compared to a day with non-laminated clouds. This can help constrain our understanding of the composition of, and precipitation processes within, laminated clouds.





## 7   Data availability

CRL data used in this paper available upon request from corresponding author (e.mccullough@dal.ca).

ECCC data is available at:

https://climate.weather.gc.ca/historical_data/search_historic_data_e.html. Parameters used are: StationID 53598, and station name "Eureka A". The data is available in CSV and XML format. The values used in this paper were last verified accessed 18 August 2019 from the ECCC website.

Note that on some of the data pages (e.g. for the 2016 data), there are three stations listed: The first two are both named "Eureka A", and the third is "Eureka Climate". The second link to "Eureka A" is the correct path to the 2016 data used in this

paper. 2017-2019 data are accessed in a similar manner.





## Appendix A:  Weather and interpretable CRL days

**Table 5.** Number of all fully interpretable days for which each type of weather was reported for at least 1h for a. 2016, b. 2017, c. 2018, and d. 2019. Note that in 2016, blowing snow occurred only on days when there was also precipitating snow, which is why the number of All Snow days matches the number of Snow days, despite the 15 days in the Blowing Snow category.

| | Laminated | Non-laminated Cloudy | Non-laminated Clear | | Laminated | Non-laminated Cloudy | Non-laminated Clear |
|---|---|---|---|---|---|---|---|
| a. 2016 | N = 61 | N = 40 | N = 27 | b. 2017 | N = 52 | N = 24 | N = 15 |
| All Precip | 43 (70 %) | 4 (10 %) | 0 (0 %) | All Precip | 41 (79 %) | 3 (13 %) | 0 (0 %) |
| All Snow | 39 (64 %) | 7 (18 %) | 1 (4 %) | All Snow | 42 (81 %) | 3 (13 %) | 0 (0 %) |
| Snow | 39 (64 %) | 3 (8 %) | 0 (0 %) | Snow | 39 (75 %) | 3 (13 %) | 0 (0 %) |
| Blowing Snow | 9 (15 %) | 6 (15 %) | 1 (4 %) | Blowing Snow | 13 (25 %) | 3 (13 %) | 0 (0 %) |
| Fog | 11 (18 %) | 4 (10 %) | 3 (10 %) | Fog | 9 (17 %) | 1 (4 %) | 0 (0 %) |
| Rain | 6 (10 %) | 1 (3 %) | 0 (0 %) | Rain | 4 (8 %) | 0 (0 %) | 0 (0 %) |
| Ice Crystals | 20 (33 %) | 11 (28 %) | 11 (41 %) | Ice Crystals | 16 (31 %) | 5 (21 %) | 1 (7 %) |
| Freezing Fog | 2 (3 %) | 0 (0 %) | 0 (0 %) | Freezing Fog | 1 (2 %) | 0 (0 %) | 0 (0 %) |
| Ice Pellets | 0 (0 %) | 0 (0 %) | 0 (0 %) | Ice Pellets | 1 (2 %) | 0 (0 %) | 0 (0 %) |
| Snow Grains | 0 (0 %) | 0 (0 %) | 0 (0 %) | Snow Grains | 1 (2 %) | 0 (0 %) | 0 (0 %) |
| c. 2018 | N = 84 | N = 52 | N = 17 | d. 2019 | N = 46 | N = 34 | N = 17 |
| All Precip | 60 (71 %) | 9 (17 %) | 0 (0 %) | All Precip | 29 (63 %) | 8 (24 %) | 1 (6 %) |
| All Snow | 57 (68 %) | 9 (17 %) | 0 (0 %) | All Snow | 32 (70 %) | 14 (41 %) | 1 (6 %) |
| Snow | 56 (67 %) | 8 (15 %) | 0 (0 %) | Snow | 27 (59 %) | 8 (24 %) | 1 (6 %) |
| Blowing Snow | 17 (20 %) | 2 (4 %) | 0 (0 %) | Blowing Snow | 16 (35 %) | 8 (24 %) | 1 (6 %) |
| Fog | 16 (19 %) | 6 (12 %) | 0 (0 %) | Fog | 13 (28 %) | 6 (18 %) | 2 (12 %) |
| Rain | 5 (6 %) | 1 (2 %) | 0 (0 %) | Rain | 3 (7 %) | 0 (0 %) | 0 (0 %) |
| Ice Crystals | 43 (51 %) | 31 (60 %) | 5 (29 %) | Ice Crystals | 23 (50 %) | 22 (65 %) | 9 (53 %) |
| Freezing Fog | 2 (2 %) | 0 (0 %) | 0 (0 %) | Freezing Fog | 0 (0 %) | 1 (3 %) | 0 (0 %) |
| Ice Pellets | 1 (1 %) | 0 (0 %) | 0 (0 %) | Ice Pellets | 0 (0 %) | 0 (0 %) | 0 (0 %) |
| Snow Grains | 1 (1 %) | 0 (0 %) | 0 (0 %) | Snow Grains | 0 (0 %) | 0 (0 %) | 0 (0 %) |





*Author contributions.* E. M. McCullough: Operation and maintenance of the lidar. Data analysis. Writing of analysis MATLAB code. Manuscript preparation. R. Wing: Contributions to statistical interpretations. Contribution to manuscript preparation. J. R. Drummond: Principal Investigator of PEARL laboratory. Contribution to manuscript preparation.

*Competing interests.* The authors declare that they have no conflict of interest.

*Acknowledgements.* This research is currently supported by the Natural Sciences and Engineering Research Council, Environment and Climate Change Canada and the Canadian Space Agency.

PEARL has been supported by a large number of agencies whose support is gratefully acknowledged: The Canadian Foundation for Innovation; the Ontario Innovation Trust; the (Ontario) Ministry of Research and Innovation; the Nova Scotia Research and Innovation Trust;

the Natural Sciences and Engineering Research Council; the Canadian Foundation for Climate and Atmospheric Science; Environment and Climate Change Canada (ECCC; who also provided the radiosonde data); Polar Continental Shelf Project; the Department of Indigenous and Northern Affairs Canada; and the Canadian Space Agency. This work was carried out during the Canadian Arctic ACE/OSIRIS Validation Campaigns of 2016, 2017, 2018, and 2019, which are funded by the Canadian Space Agency, Environment and Climate Change Canada, the Natural Sciences and Engineering Research Council of Canada, and the Northern Scientific Training Program. This particular project has also been supported by NSERC Discovery Grants and Northern Supplement Grants held by James R. Drummond, Robert J. Sica, and Kaley A. Walker, the NSERC CREATE Training Program in Arctic Atmospheric Science (PI: Kim Strong). Thomas J. Duck was the initial

Principal Investigator of CRL lidar, and contributed helpful comments regarding early drafts of the paper.

In addition, the authors thank the following groups and individuals for their support during field campaigns at Eureka: PEARL site manager Pierre Fogal; Canadian Arctic ACE/OSIRIS Validation Campaign project lead Kaley A. Walker, CRL operators: Graeme Nott, Chris Perro, Colin P. Thackray, Jason Hopper, Shayamila Mahagammulla Gamage, Jon Doyle; Canadian Network for the Detection of Atmospheric Change (CANDAC) support scientist Alexey Tikhomirov; CANDAC operators: Mike Maurice, Peter McGovern, John Gallagher, Alexei

Khmel, Paul Leowen, Ashley Harrett, Keith MacQuarrie, Oleg Mikhailov, and Matt Okraszewski; and the Eureka Weather Station staff. Thanks to members of CANDAC for their helpful discussions.





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
