# Peer review of "Finely laminated Arctic mixed-phase clouds occur frequently and are correlated with snow"

_Atmospheric Chemistry and Physics, 2020_

## Referee Comment (RC1) · Anonymous Referee #1 · 10 Apr 2020

Review of "Finely laminated Artic mixed-phase clouds occur frequently and are correlated with snow" by McCullough, Wing, and Drummond.

This study by McCullough et al. is one of few studies presenting very high-resolution observations of clouds. This study reveals that laminated features in Artic clouds are not uncommon and attempts to quantify the relative occurrence of this phenomenon. It also estimates correlations between the occurrence of laminated clouds and the occurrence of precipitation, which could help improve our understanding of the formation mechanism of these laminations and/or of the impact of these laminations on precipitation.

[Figure]

Although I believe answers to these questions would make a great contribution to the field of atmospheric science, I have several issues with this manuscript as it now stands. For reasons detailed below, I would recommend this manuscript be rejected, but I would encourage the authors to resubmit.

————————- Highlights ————————-

-Science

This study is one of few studies presenting very high-resolution observations of clouds.

-Figures

The figure presented through the manuscript are impeccable. The authors have used appropriate font size, color contrast, labels, and legends.

————————- Major comments ————————-

- Lack of precision in the identification of laminated clouds and in the use of certain terminology

This manuscript relies on manual inspection of plots to create a climatology of the occurrence of laminated clouds. Manual inspection is a highly subjective way to classify observed scene and an impossible one to reproduce. I believe it is imperative for published science to be 100% reproducible. I would recommend the authors start from the vague rules they provide in P 5 L 15-19 to create a precise, programmable set of rules defining what are laminated clouds. As a last resort, I would ask the authors to provide all figures as supplemental material each one labelled according to their scene classification. I believe "Available upon request from the corresponding authors" is simply not sufficient in this case.

If the rules defined in P 5 L 15-19 were more precisely defined and implemented I would agree they could be appropriate to identify laminated clouds. That being said, they would not be sufficient to identify mixed-phased or multi-layer conditions which the

authors claim to study as stated in the manuscript and in the title.

Mixed-phase: According to work by Shupe and others, additional information besides photon count is helpful to assess cloud microphysical phase. Thus, the authors statements throughout the manuscript and in the title that "the clouds observed over this 3.5-year climatology are mixed-phase" is not strongly supported. This work would benefit for example from using depolarization information.

Multi-layer: Multi-layer clouds are generally defined as clouds containing multiple liquid layers. Given this, to sustain their claim that the statistics presented in this study pertain to multi-layer clouds, the authors should probably perform a phase classification to distinguish between liquid and ice which both can produce high photon counts. Moreover, I would argue that the types of cloud presented in Fig. 3a and in Fig. 3d are quite different yet the authors consider them together in their statistics. At first glance, I would label the cloud in Fig. 3d as a "traditional multi-layer clouds", and I would certainly need to be convinced that the cloud in Fig. 3a is a "multi-layer" clouds. Of course, this study would also be valuable if it was simply describing clouds in general (i.e., without multi-layer statements). That being said, this approach would require a rewriting of the introduction which claims that part of the uniqueness of this study is that it focuses on multi-layer clouds.

- Statistical bias caused by the scene classification methodology

This study is based on the analysis of entire days (i.e., 24-hrs) of observations. I believe this was done to keep data size manageable for manual inspection. This however creates an issue related to data gaps. The authors attempt to address this issue by defining "interpretable" days. That being said, they do not use this method consistently. For instance, as this classification now stands, a day with only 30 min of laminated clouds is interpretable even if 23.5 hrs of data are missing, but a day with 22 hrs of non-laminated clouds and 2 hours of missing data is non-interpretable. This methodology is sure to make all relative statistics presented in the current study biased high toward

laminated conditions. I believe it would be fairer first to remove days with > 1-hr of missing data, then days with low-level cloud obstruction. I would consider the rest of the scenes as interpretable. Then I would classify those as clear or cloudy and I would further classify the cloudy ones as laminated and non-laminated. This would ensure that laminated and non-laminated conditions are estimated using the same sample size of "interpretable" cases and would generate unbiased relative frequency of occurrences. I would also recommend that the authors use 1-h scenes rather than 24-h scenes. This would correspond better with the time resolution of the weather reports and would likely increase the number of interpretable scenes.

————————- Minor comments ————————-

- Abstract:

The abstract could be written in such a way as to be much more insightful. For example, "P 1 L 10-11" would be more informative if some actual correlation coefficients were given. Also, it would be more informative if information about part II of investigation A was provided; for instance, are there notable monthly differences in the occurrence of laminated clouds?

P1 L 2-4: This sentence is very long. Please consider rewriting it. P 1 L 6:" the expression "interpretable days" is not defined in the abstract thus creating confusion for anyone who has yet to read the complete manuscript.

- Number of tables

Have the authors consider putting some of their tables in an appendix or perhaps submitting some of them as supplemental material?

- Introduction

I would encourage the authors to shorten their introduction and to be more focused on what makes their study unique which is the fact that they provide very high-resolution observations of clouds and put them in context with precipitation occurrence. For example, I would remove P 1 L 15-17, L 23-25, P 2 L-1-4, and in particular L 8-13 (Anyway "measurements which parameterize" is an incorrect statement since measurements do not "parameterize" they are "used to evaluate" or "used to construct parameterizations").

I think your best statement in the introduction is P 3 L 3-5

I think the introduction would benefit from more background information on previous studies focused on high-resolution observations of mixed-phased clouds such as those conducted by Verlinde and coauthors.

- Organization:

Figure 1 is presented in the introduction before any information has been provided about the sensor used to record the information presented. I encourage the authors to move this figure after or within the methods section.

- Spelling and grammar:

There are several spelling and grammar errors throughout the manuscript. For one, the word "occurrence", which is written at least three different ways: "occurance", "occurrance", and "occurence". I would encourage the authors to run a spell check before resubmitting their manuscript.

---

## Referee Comment (RC2) · Anonymous Referee #2 · 14 Apr 2020

General Comment: Arctic mixed-phase clouds play an important role in the Earth's energy and water budgets. However, its morphology is complex, and the occurrence of each cloud type is still unclear. This manuscript focuses on laminated mixed-phase clouds and its relationships with weather conditions using lidar data from multi-year observations. The manuscript is well organized, and figures and tables are very clear. The dataset and analysis technique are unique and interesting. Although the analysis method is well described and the limitations of the technique are considered the discussion, I have a few concerns in the method and definition, which might affect the conclusions. The concerns below should be addressed before the manuscript is accepted for publication.

[Figure]

Major comments

1. I have questions and concerns about definitions of the "laminated clouds" focused in this study. The manuscript stated "we focus on a different type of mixed-phase cloud which contains layers of ice and liquid throughout the volume of the cloud." Is this same as "ice (precipitating) cloud in which more than one liquid layers were embedded"? Because "Arctic mixed-phase clouds" implies that ice particles and liquid particles coexist and temperatures are below the freezing temperature in the cloud depth, so the liquid particles can be supercooled liquid droplets. However, the analysis of the study seems to also include rain precipitating clouds, where the temperature could be greater than $0°C$. Please give more detailed descriptions of the target clouds in terms of this and the temperature information of the target cases.

2. I have a question about the third criterion for "Laminated." Based on the criterion, the laminated cloud can include three thin cloud layers (i.e. quasi-horizontal stripes) in the 75 m depth. I felt that this is too narrow. If the depth between the first layer and the third layer exceeded 75 m, was the cloud not included? Why was this criterion used to define laminated clouds? What kind of clouds did you want to include/exclude by this criterion?

3. What was the thickness of the "quasi-horizontal stripes"? How did you define the quasi-horizontal stripes (i.e. what is the vertical gradient of the lidar backscatter)? Was this definition same as one by O'Connor's et al. (2004) for ceilometer' cloud base height? O'Connor, Ewan J., Anthony J. Illingworth, and Robin J. Hogan. "A Technique for Autocalibration of Cloud Lidar." Journal of Atmospheric & Oceanic Technology 21, no. 5 (2004).

4. I have a concern about the way to count laminated cloud day. If my understand was correct, the laminated cloud day was counted when the laminated condition lasted 0.5 h at least within 24 hours. Based on this definition, it is possible that the most of time could be non-laminated condition in a day that was classified as "laminated day". Is

this reasonable to select laminated days? My concern is that if "non laminated day" was counted using a definition that non-laminated condition was lasted >0.5 h, many laminated days selected in the manuscript could be counted as "non-laminated day." In that case, the results obtained in the present study could be opposite. Is the conclusion also affected by the criterion and selection?

5. While the minimum duration of laminated condition was 0.5 h, the weather condition used in the study was based on the daily reports which provide only a few condition categories per day. How did you ensure the correlation of these different time resolution data?

6. Based on the classification method, I think that the "Undetermined (obscured)" category implies a low-level thick mixed-phased cloud. I am curious about this category and wondering if this category was included in Investigation B, how this category was correlated with the weather conditions.

7. On p. 13, Pearson's r correlation analysis: Please discuss the variability of r values over the years. What is the reason of the variability?

8. Discussion for Investigation A: Please more discuss about the seasonality of the occurrence in terms of meteorological conditions, environment, etc.. Figure 5 shows clear seasonal variability in each year except 2017; there is a peak at May and April. What is the reason of the seasonal variability? Why does 2017 show different variability from other years?

9. Lines 10-11 on p. 17: Same as comment #4, the laminated cloud case was identified when the laminated condition lasted only for 0.5 h. The most of time might be non-laminated condition. If "non-laminated cloud day" was counted as a day where non-laminated condition lasted for >0.5 h, the result could say "non laminated clouds may form an even larger component of the atmosphere at Eureka." Please carefully mention/discuss considering the limitation of the analysis technique. I have the same comment a statement on lines 18-19 on p.18.

10. Please also discuss about the seasonal variabilities in terms of meteorological conditions, environment. etc.

Minor comments

1. Lines 21-22 on p.13: The occurrences in November and December show large variability in Figure 5. Therefore, this sentence can mislead the readers. This sentence should be rephrased carefully.

2. Figure1: Please specify the laminated cloud regions identified by the definitions.

3. Line 10 on p. 13: Remove "is."

---

## Author Comment (AC1) · 1 Aug 2020

Please refer to supplementary PDF file for responses.

Please also note the supplement to this comment:
https://www.atmos-chem-phys-discuss.net/acp-2020-186/acp-2020-186-AC1-supplement.pdf
* * *